Characteristics of gauged abrupt wave fronts (walls of water) in flash floods in Scotland

David R. Archer<sup>1, 2</sup> Felipe Fileni<sup>1</sup>, Sam A. Watkiss<sup>3</sup>, Hayley J Fowler<sup>1</sup>

<sup>1</sup>School of Engineering, Newcastle University, Newcastle upon Tyne, NE1 7RU, United Kingdom

<sup>2</sup>JBA Trust, North Yorks, UK

<sup>3</sup>JBA Consulting, Skipton, North Yorks, UK

Correspondence to: David Archer (d.r.archer@ncl.ac.uk)

Abstract.

Extremely rapid rates of rise in river level and discharge are a subset of flash floods ('abrupt wave front floods', AWFs) and

are separate hazards from peak river level. They pose a danger to life to river users and occur mainly in the summer. Using

level and discharge records from 260 Scottish gauged catchments, we present the spatial distribution of annual maximum 15

minute rises in river level and discharge, along with derived metrics to assess the severity of AWF events. These include

normalised and proportional measures of flow change, as well as ratios that characterise the intensity of AWF events. We

estimate wave celerity by analysing the time difference in wave onset recorded successive gauging stations along a river

channel. This approach is applied to several AWF events on the River Findhorn in northeast Scotland, allowing for detailed

examination of their dynamics. Our findings suggest that flood forecasting models with outputs of peak discharge and river

level, may not adequately represent the risk posed by rapidly rising flows, especially at national scales where hydroclimatic

and geomorphological variability trigger different AWF metrics. We show that AWFs may intensify downstream, with wave

fronts steepening as they travel through lowland river reaches, as observed in multiple events on the River Findhorn, showing

a necessity of more accurate and frequent river measurements. We conclude that AWFs need better monitoring forecasting

and warning, particularly as extreme downpours are becoming more frequent with global warming.

Keywords: Flash flood, Abrupt wave front, kinematic shock, Scotland

1 Introduction

Extremely rapid rates of rise in river level and velocity, often described as 'walls of water', are a subset of flash floods (also

called 'abrupt wave front floods', AWFs) (Archer and Fowler, 2018). They are separate hazards from peak levels whose

principal impact is on the flooding of property and economic loss. The rapidity of onset of AWFs, often as a visible wave,

provides a critical danger to the lives of river users such as anglers and swimmers even when the peak river level is not severe.

On a worldwide basis there is a growing recognition of the hazard of floods with a very rapid rate of rise. Collischonn & Kobiyama (2019) noted seven events in southern Brazil in the period 2008 to 2019 in which a total of 16 people were washed away and drowned. Viggiani (2020) compiled a list of 19 'surge waves' from around the world which caused significant loss of life. Here we examine gauged records of such events in Scotland.

Scotland is subject to river flooding from several driving forces. As a mountainous country on the Atlantic fringe, it suffers flooding from persistent, orographically enhanced frontal and cyclonic rainfall and from melting snow (SEPA, 2022). Convective activity, the source of flash floods, is much weaker in Scotland than in southern England and adjacent continent (Hayward et al., 2022). Nevertheless, intense summer convective rainfall has historically caused serious flash floods in Scotland both from surface water in urban areas and river flooding. In compiling a historical chronology of flash floods in Britain from 1700, Archer and Fowler (2021) listed 612 events in Scotland of a national total of 7921

(https://www.jbatrust.org/about-the-jba-trust/how-we-help/publications-resources/rivers-and-coasts/british-chronology-of-flash-floods/). Of these 43 were identified as abrupt wave front floods (AWFs) in rivers from observers' descriptions as 'walls of water' or implied by impact including loss of life by drowning and bridge destruction. The identification of historical AWFs prompted investigation of rapid rates of rise in gauged records of level and flow. Archer et al., 2024 examined the occurrence of such AWFs in northern England, noting their occurrence on every major catchment draining the Pennines. We use lessons learned from this analysis in the extension here to neighbouring Scotland.

AWFs are usually generated in steep upstream tributaries but may be transmitted downstream over tens of kilometres. A gauged example from the Pennines (Archer and Fowler, 2018) shows that the flood wave may steepen as it progresses downstream (Fig. 1). At the upstream station at Alston (118 km²) there is a gradual rise of 62 m³s⁻¹ in an hour before a sharp 15 min rise of 117 m³s⁻¹ followed by a 15 min rise of 80 m³s⁻¹ At the next downstream gauge of Featherstone (322 km²), the initial gradual increase in flow at Alston has been absorbed and discharge rises abruptly from 2 m³s⁻¹ to 168 m³s⁻¹ within 15 minutes (Fig 1a). The hazard to river users is much lower at Alston where the progressive initial rise provides a greater opportunity to escape than at Featherstone. The steep wave front continued downstream and absorbed an early tributary inflow from the River Allen between gauging stations at Haydon Bridge (751 km²) on the South Tyne and Bywell (2176 km²) on the main Tyne (Fig1b). Both gauged and historical observations at a single location may not therefore represent the most severe hazard experienced in a flood event.

The hazard of abrupt wave fronts is a combination of the simultaneous increase in level and velocity. Gauged observations on the River Tyne show that velocity can increase from an initial value of less than 0.5m s<sup>-1</sup> to more than 3.0 m s<sup>-1</sup> within the same time interval (Archer and Fowler, 2024). Collischonn & Oliveira, 2023 note the celerity of a flood wave on the River Luthern in Switzerland in May 2023 and recorded the arrival time of an AWF at 3 different points. From this information the flood wave celerity was calculated as around 3.7 m s<sup>-1</sup>. Since velocity is rarely measured in flood events and discharge is estimated from observed level via a rating curve the assessment of mean or maximum velocity during AWFs at the gauging section is difficult. We therefore use a variety of measures of 15 min change in discharge using Scottish gauged data as a means of assessing the severity and rarity of AWFs for a single catchment and as a means of comparison between catchments.

Figure 1. a. The progress of an AWF on the South Tyne catchments on 30 July 2002 illustrating the downstream absorption of an initial gradual rise, and b. The absorption of a tributary inflow from the River Allen, between Haydon Bridge on the South Tyne and Bywell on the main Tyne. Note that the peak discharge changes little over the 72 km reach between Alston and Bywell.

This study presents novel large-sample evidence of abrupt wave front floods (AWFs) in Scottish rivers, offering new perspectives on their spatial distribution, downstream evolution, and hazard potential. Here, we place these results in the context of hydrodynamic theory and risk analysis. After identifying key data and methodological limitations, we propose future directions for improving the detection and understanding of AWFs.

## 2 Data



The 15 min flow and level dataset used was sourced from the Scottish Environment Protection Agency (SEPA) time series data service (API). The website has 390 level and 315 flow stations available, with more than 20000 years of data in total (Fileni et al., 2023). The data provided by SEPA are based on 15 min measurements of river level and these are with few exceptions converted to flow by rating equations derived from individual discharge measurements at given levels combined with weir equations. SEPA hydrometry team reviews level measurements monthly and rating curves annually. These reviews result in the correction of data artefacts before publishing the timeseries and necessary changes in the rating curves and flow conversions

For the study 260 stations were selected: these correspond to ones that present both flow/level data and a National River Flow Archive (NRFA) identifier (https://nrfa.ceh.ac.uk/data/search). The records provided a median length of 33 years with the earliest records dating back to the 1950s (Fig. 2). The rates of rise in level and discharge were computed by calculating the

first derivative for every timestep of the timeseries, from which the annual maximum values of rise in level (HW15) and discharge (QW15) were extracted.

Figure 2: The 260 gauging stations in Scotland that were used for the study. Record length at each station is represented by the circle colour and catchment size represented by circle size.

From the annual maxima, the five highest rises for the months of April to September were selected, as this is the period when convective storms produce sufficiently intense rainfall in Scotland to generate AWFs. These hydrographs were then visually inspected to validate each event as an AWF, following a comprehensive QC procedure (Fileni et al. 2023). Some events were

eliminated as spurious spikes or otherwise inconsistent hydrological behaviour in the record; others were excluded as part of a 'normal' flood resulting from persistent heavy rainfall and usually near to the upper end of a rising limb rather than rising rapidly from low flow. Coincidence between level and flow station maximum rates of rise, where the maximum HW15<sub>abs</sub> (Eq 1) exceeded 0.6 m, occurred for 48% of stations. This variation between level and flow maxima can be attributed to the logarithmic relationship between level and discharge so that a given level change results in a higher flow change at a higher starting point.

#### 3 Methods





#### 3.1 Change in 15 min river level

Increase in river level and velocity combine to create the hazard to river users during AWFs. Previous analyses, especially when applied to multiple stations, has focused on changing level as the most obvious and visible feature of an AWF. We continue to use HW15<sub>abs</sub> (Eq. 1) for this study of Scottish AWFs. 'Peak' here refers to the upper limit of the 15 min rise.

$$HW15_{abs} = HW_{max peak} - HW_{max peak-1}$$
 Eq 1

For an individual event, Archer (1994) and Archer et al. (2017) used an annual maximum series of 15 min rise in level (HW15) to estimate the return period of an extreme rise on the River Wansbeck in northeast England. Assuming a generalised logistic distribution for gauged data only, the return period of the 1994 15 min rate of rise of 1.26 m was calculated as 140 years but reduced to 60 years when historical precedents beyond the digital record were considered. With the availability of annual maximum rate of rise statistics for Scotland, it is possible to apply flood frequency analysis to all stations. For analysis of events in the Pennines a simpler metric of the ratio of the absolute maximum to the observed median for each station was used (Archer et al., 2024) to assess the severity for an individual catchment. For example, on the South Tyne the median maximum 15 min rise, HW<sub>med</sub> at Haydon Bridge is 0.70 m and the HW15<sub>abs</sub> is 1.49 m (a ratio of 2.1) compared with the River Wansbeck where the median maximum 15 min rise is 0.28 m and the HW15<sub>abs</sub> is 1.26 m (a ratio of 4.5). This shows that for rivers with the greatest propensity for AWFs to occur (e.g. South Tyne) the ratio may be smaller than on those (e.g. Wansbeck) where such events are rare. The hazard for river users may thus be greater on rivers where such events are least expected. In this study of Scottish rivers, we focus on the ratio of absolute maximum 15 min rise in flow to the median, rather than increase in level.

#### 3.2 Change in velocity

Mean or maximum velocity in a cross section during an AWF is a key component of the hazard but is difficult to measure or assess. Use of in-river measurement is impractical owing to bedload and heavy floating debris. Measurement of surface

velocity may be achieved by methods using fixed cameras, but drone photography may be precluded by the time taken to reach a site.

Initial velocity before the arrival of the AWF is low (a condition of the transmission of a kinematic wave) and the velocity at a station for the duration of the AWF is likely to be dominated by wave celerity. Celerity can be determined using the arrival time of a wave front at multiple sites with known distances between them. This method was applied to a single catchment in Scotland, the River Findhorn.

# 3.3 Maximum change in 15 min discharge





- In the absence of velocity estimates, several aspects of discharge measurement are used to compare the severity of rapid rates of rise within and between catchments.
  - 1. The maximum absolute increase in discharge between the beginning and end of the 15 min period based on the standard rating curve, QW15<sub>Abs</sub> (Eq. 2). A given increase will have greater impact on a small catchment with a narrow and confined cross-section but, for practical purposes, we have excluded most events and catchments where the increase is less than 10 m<sup>3</sup> s<sup>-1</sup>, except where the rise in level is greater than 0.4 m.
  - 2. The rate of rise normalized by the median annual maxima peak flow (QMED), described by QW15<sub>Qmed</sub> (Eq. 3). Normalizing the data by QMED facilitates inter-catchment comparisons of severity of flow increase, independent of catchment characteristics especially of size. The alternative of normalising by catchment area has been used by Amengual (2025) as a means of characterising extreme flash floods in Mediterranean Spain. Although AWFs are usually generated on only a small area of a catchment, there is the potential for larger catchments to generate larger flows where the flow from incoming tributaries is combined.
  - 3. The ratio of maximum to median 15 min annual maximum rise, QW15<sub>Ratio</sub> (Eq. 4), provides a measure of the comparative severity of the most extreme AWF within a catchment. In a similar fashion to QMED (the median annual maximum peak flow), the median annual maxima rate of rise was calculated (QW15<sub>med</sub>). This metric then estimates the frequency of occurrence of AWFs by dividing the AWF absolute value by QW15<sub>med</sub> (Eq. 3).
  - 4. The proportional increase in flow from the initial flow to the peak of the 15 min rise, QW15<sub>Prop</sub> (Eq. 5). This is a measure of the magnitude of the change in a 15 min period and is an important contributor to the hazard. However, the measure may be biased when the river is initially dry (when the measure is infinite) or when the flow is very low. To avoid infinity values and to compute only relevant relative increases, the relative rate of rise was computed solely when the final timestep exceeded the 10<sup>th</sup> percentile flow.

$$\begin{aligned} & \text{QW15}_{\text{abs}} = \text{QW15}_{\text{max peak}} - \text{QW15}_{\text{max peak}-1} & \text{Eq 2} \\ & \text{QW15}_{\text{QMED}} = \frac{\text{QW15}_{\text{abs}}}{\text{QMED}} & \text{Eq 3} \\ & \text{QW15}_{\text{Ratio}} = \frac{\text{QW15}_{\text{abs}}}{\text{OW15med}} & \text{Eq 4} \end{aligned}$$

$$QW15_{prop} = \frac{QW15_{abs peak}}{QW15_{abs peak-1}}$$
 Eq 5

Significant AWF events were found on 93 catchments (out of 260)

#### 4 Results





Results are presented as a series of maps of Scotland for each of the measures of level or discharge as follows:

- 1. Absolute maximum change in 15 min river level (HW15<sub>abs</sub>)
- 2. Absolute maximum change in 15 min discharge (QW15<sub>abs</sub>)
  - 3. 15 min rate of rise (QW15<sub>abs</sub>) normalized by the median annual maxima peak flow (QMED)
  - 4. The ratio of absolute maximum change in 15 min discharge (QW15<sub>abs</sub>) to median maximum 15 min rise (QW15<sub>med</sub>)
  - 5. The proportional increase in flow from the initial flow to the peak of the 15 min rise (QW15<sub>Prop</sub>)

## 4.1 Change in 15 min river level – HW15<sub>abs</sub>

For the purposes of identifying AWFs, our analysis has been restricted to the summer months of April to September where events are generated by intense, often localised, convective rainfall. Rapid increases in level also occur during the winter months at many stations resulting from persistent and widespread heavy rainfall. The maximum 15 min rise in level or discharge in winter events usually occurs as part of the rising limb of a normal hydrograph and provides much less risk to river users. However, it is possible that we have missed some AWFs outside of the summer period.

The geographical distribution of events, the magnitude of the largest event, and the number of stations in each range of maximum 15 min level change at each station is shown in Fig. 3a. AWFs have been observed over most of the country but with perhaps the greatest concentrations in the rivers of the northeast, including the very high HW15<sub>abs</sub> on the River Findhorn at Forres (1.87 m). Fewer events have been observed on rivers in the western Highlands; in the central lowlands and on the southern fringe of the mountains the magnitude of AWF events is smaller than elsewhere. AWFs are rare on the main stem of rivers with upstream lakes and reservoirs, such as the River Tay, although they may occur on upstream tributaries. AWFs usually originate on steep upland tributaries but there are few gauging stations near to the point of generation. The median catchment area where events were observed is 201 km² but they range in area up to 2,861 km² for the River Spey at Boat o Brig. Only 10 stations (10.8%) with AWFs are under 50 km², where such events are typically generated, which may reflect the fact that many small catchments are ungauged. The average elevation of gauging stations is less than 50 m asl and 16 of 93 stations (17%) are below 10 m asl, including the Findhorn at Forres with a catchment area of 782 km². At many stations only a single event with a rise greater than 0.40 m in 15m was observed. However, five such events occurred on Ruchill Water at Cultybraggan, where Cranston and Black (2006) previously noted the short lead times of floods but not the rapid rate of rise; the largest event had a 15 min rise of 1.88 m on a catchment area of just 99.5 km²

We note that level is not a completely reliable measure for comparison between stations, since increase in level depends on the stage/discharge relationship and the configuration of the control section, whether natural or constructed, at each station. With respect to natural channels, Wharton (1995) notes that for British rivers there is a strong relationship between channel width or cross-sectional area and river flood discharge, especially for flows confined within a channel. However, we suggest that other measures of severity are necessary for increased understanding of AWFs.

Figure 3a. The maximum change in 15 min level at Scottish stations with AWFs (HW15<sub>abs</sub>) and the number of stations in each range, and b. The maximum change in 15 min discharge (QW15<sub>abs</sub>) showing comparative magnitudes and the number of stations in each range.

## 4.2 Change in 15 min discharge - QW15abs

The geographical distribution and magnitude of the largest QW15<sub>abs</sub> at each station and the distribution of values is shown in Fig 3b. The comparative magnitude of level and discharge may vary, especially on catchments of differing size. Thus, the Avon at Delnashaugh (catchment area 543 km²) and the large Spey catchment at Boat o Brig (2861 km²) have events of similar QW15<sub>abs</sub> magnitude of 144 m³ s⁻¹ and 152 m³ s⁻¹ respectively but a differing HW15<sub>abs</sub> of 1.47 m and 0.70 m. These differences reflect the greater channel capacity of the larger river. Conversely, stations with a similar HW15<sub>abs</sub> may have a different

QW15<sub>abs</sub>. Thus, Ettrick Water at Brockhope (37.5 km<sup>2</sup>) and the Esk at Canonbie (495 km<sup>2</sup>) in southern Scotland have events of a similar HW15<sub>abs</sub> of 1.30 m and 1.36 m but very different QW15<sub>abs</sub> of 42 m<sup>3</sup> s<sup>-1</sup> and 130 m<sup>3</sup> s<sup>-1</sup> respectively.

The hydrograph for the Ruchill Water at Cultybraggan (Fig. 4a) is typical of AWFs in Scotland, with a very rapid initial rise from a very low flow followed by the peak discharge less than an hour later and a rapid recession, returning to a low flow within 12 hours; the HW15<sub>abs</sub> of 1.88 m for this event was the highest observed in Scotland. The transition from rising limb to peak is even more pronounced for the events shown on the River Avon at Delnashaugh (Fig. 4b) and the River Dee at Polhollick (Fig 4c)

Fig. 4: Hydrograph of the QW15<sub>abs</sub> AWF on: a. the Ruchill Water at Cultybraggan on 4 August 2012; b. the River Avon at Delnashaugh on 7 June 1980; c. River Dee at Polhollick on 22 September 1991. In each case the 15 min change in flow is shown.

# 4.3 Rate of rise normalized by the median annual maximum peak flow (QMED) - QW15QMED

Comparison of the severity of an AWF between catchments is constrained by the influence of other catchment characteristics which influence the magnitude of floods, notably the influence of catchment area, as noted above. However, area is not the only factor and another measure of catchment susceptibility, QMED (the median annual peak flood), has been used to normalise the hazard of flood discharge between catchments. Normalised values are mapped for Scottish catchments in Fig. 5a and the distribution of values of the ratio is shown.

Although QW15<sub>abs</sub> is a high proportion of the peak flow in AWF events, as demonstrated in Figs. 2b and 4a and b, it is a modest proportion of QMED. The median value is 0.36 on stations that are prone to AWFs, and only two stations have values >0.8. For example, in the River Strontian at Ariondle (25.2 km²) the largest QW15<sub>abs</sub> exceeded QMED (QW15<sub>asd</sub>/QMED = 1.48) but at the same time its ratio of maximum to median (Sect 4.4) (QW15<sub>abs</sub>/QW15<sub>med</sub>) is the lowest in the dataset at 1.48, suggesting that AWFs at this station are both frequent and severe. In contrast, for the River Nethan at Kirkmuirhill where QW15<sub>abs</sub>/QMED =0.98, the maximum to median (QW15<sub>abs</sub>/QW15<sub>med</sub>) of 6.4 suggests that the event of 4 July 2001 for this station was very unusual. For large catchments such as the Spey at Boat o Brig (2861 km²) and the Dee at Woodend (1370 km²) where actual QW15<sub>abs</sub> values were high, QW15<sub>abs</sub>/QMED were not exceptional (<0.3). However, some stations displaying the largest HW15<sub>abs</sub> and QW15<sub>abs</sub> also had very high QW15<sub>abs</sub>/QMED, for example the Avon at Dalnashaugh (0.68) and the Ruchill Water at Cultybraggan (0.67). This indicates that AWFs here had an extreme severity, both with respect to their own catchment and when compared across catchments.

#### 4.4 Ratio of maximum to median 15 min rise in discharge-QW15Ratio

The ratio of the maximum to the median 15 min rise in discharge is a simple measure of the severity of the most extreme event on a catchment and is thus a measure of the additional hazard provided by an AWF. This ratio is mapped for Scottish catchments, and the distribution of values is shown in Fig. 5b.

For Scottish gauges the median ratio, on stations that are prone to AWFs, was found to be 2.9, but the most extreme ratios (>5.0) were experienced on catchments where the actual maximum level or discharge rise was not extreme. For example, the River Livet at Minmore with a catchment area of 104 km<sup>2</sup> and a maximum 15 min rise of level and discharge of 0.73 m and 22.3 m<sup>3</sup> s<sup>-1</sup>, had a ratio of 8.0. The River Nethan at Kirkmuirhill (66 km<sup>2</sup>) with a 15 min rise in level and discharge of 1.05 m and 34.7 m<sup>3</sup> s<sup>-1</sup>, had a ratio of 6.4. No catchment with a maximum rise > 1.0m and >100 m<sup>3</sup> s<sup>-1</sup> had a ratio greater than 3.8.

Fig. 5a. Maximum 15 min rise in flow normalised by QMED and the number of stations at which the range of values of the ratio of QW15<sub>abs</sub> to QMED occurred and b. Ratio of the absolute maximum 15 min rise in flow (QW15<sub>abs</sub>) to the median rise (QW15<sub>med</sub>) and the number of stations for which the range of values of the ratio occurred.

#### 4.5 Proportional increase in flow-QW15<sub>Prop</sub>

A key feature of risk for river users is the proportional increase in flow from the initial discharge to the magnitude of the AWF as assessed at the end of the 15 min observation interval. Some of these values can be very high and theoretically infinite if the initial channel is dry (but then even more reason to be a hazard!). Values are mapped for Scottish catchments, and the distribution of values is shown in Fig. 6.

Fig. 6: The proportional increase from the initial flow to the end of the 15 min maximum rise and the number of stations at which the range of values of the proportional increase in flow was experienced.

The median value, on stations that are prone to AWFs, was found to be 10 times the initial flow but of the 11 stations with an increase of 30 times, the Ettrick Water at Brockhope (37.5 km²) had an increase of more than 100 times. The smallest gauged catchments were generally those with the largest increases but an exception is the River Findhorn at Shenachie (416 km²) with a proportional rise of 43. Large catchments such as the Spey, Dee and Don had proportional rises of less than 10.

#### 4.6 Estimation of flood celerity - River Findhorn



Wave celerity is the primary component of the perceived velocity at a station during an AWF; examples show that the initial velocity before the arrival of an AWF is low (a condition of the transmission of a kinematic wave). Collischonn and Oliviera (2023) give an example of the timing of a visible wave front between two points on the Luthern River in Switzerland where

they calculate a wave celerity of 3.7 m s<sup>-1</sup> along a 5 km reach. For the event of July 2002 on the River Tyne, the wave celerity between the upper stations of Alston and Featherstone was 3.6 m s<sup>-1</sup> over a 16.3 km reach and 3.1 m s<sup>-1</sup> for the lower 33.4 km reach between Haydon Bridge and Bywell. In either case, an increase from an initial velocity of less than 0.5 m s<sup>-1</sup> in 15 minutes or less would pose a serious risk to life to anglers, canoeists, and swimmers.


The River Findhorn in northeast Scotland has a long narrow steep-sided catchment, rising in the Monadhliath Mountains with its highest point at 945 m ASL. Bedrock is predominantly metamorphic, with an extensive blanket peat moorland and minimal tree cover except in the lowest reaches. It is gauged at two points: on the main stem at Shenachie (catchment area 416 km² and station elevation 252m ASL) and at Forres (catchment area 782 km² and station elevation 11 m ASL). The river distance from Shenachie to Forres is 49 km. There is one significant gauged tributary, the River Divie, gauged at Dunphail (catchment area 165 km² and station elevation 117 m ASL) which joins the main stem at approximately 18 km upstream from Forres.

The gauging stations at Shenachie and Forres have long digital records with a start date for digital records at Shenachie in 1961, at Forres in 1959 and at Dunphail in 1982. Several events in the record show evidence of major AWFs at one or both main stem stations. Timing of wave front and peak with the distance can be used to assess celerity over the reach and provide estimates of the celerity at Forres.

Fig. 7: Hydrographs of flow and 15 min rate of change at Shenachie and Forres for the 17 July 1980 event.

Figure 7 shows an already established AWF at Shenachie with a 15 min rise of 124 m<sup>3</sup> s<sup>-1</sup>, progressing to an even steeper rise of 156 m<sup>3</sup> s<sup>-1</sup> at Forres. There is clearly a problem with discharge estimation at one or both stations, with a decreasing flow volume downstream, but timings are expected to be correct. With a rise time of the wave front between the stations of 4.5 hours, the average celerity over the reach is 3.02 m sec<sup>-1</sup>. However, the downstream hydrograph seems compressed so that the travel time of 4 hours for the peak is less than that of the wave front, giving a celerity of 3.40 m s<sup>-1</sup>. Similar events occurred on 1 September 2005 and 17 August 2014, with average wave front celerities of 2.86 m s<sup>-1</sup> and 3.02 m s<sup>-1</sup> respectively. Peak travel times and celerities were 3.40 m s<sup>-1</sup>, and 2.72 m s<sup>-1</sup> for the more complex hydrograph of 17 August 2014.

The flood of 7 Jun 1990 shows a more remarkable transformation within the reach. At Shenachie there is a normal hydrograph, with a steady rise to peak and the highest 15 min rise of 0.39 m in the middle of the rising limb. However, at Forres, the water level rose suddenly from the low level of 0.3 m to 2.17 m and discharge rose from 7 m<sup>3</sup> s<sup>-1</sup> to 216 m<sup>3</sup> s<sup>-1</sup> in 15 minutes then continued to rise at a slower rate for a further 3 hours to peak at 529 m<sup>3</sup> s<sup>-1</sup>. This event is the largest observed QW15<sub>abs</sub> in Scotland. In each of these events, the flow in the River Divie remained below 10 m<sup>3</sup> s<sup>-1</sup>. Given the absence of a defined upstream wave front it was not possible to assess the celerity in the reach. This flood provided the annual maximum peak flow and was rank 8 in a 64-year record, yet still far short of the maximum gauged peak flow of 1,021 m<sup>3</sup> s<sup>-1</sup> on 17 August 1970, and an estimated 1,484 m<sup>3</sup> s<sup>-1</sup> for the 'Muckle spate' of 1829 at a point upstream on the Findhorn (McEwan & Werritty, 2007).




Fig. 8: Hydrographs of flow and 15 min rate of change at Shenachie and Forres for the 7 June 1990 event.

With the increasing wave front magnitude as it progresses downstream, it is probable that the wave accelerated to a celerity greater than the average for the reach as it approached the Forres gauging station. We suggest that it therefore would have posed a very serious threat to river users.

## 5.Discussion







## 5.1 Integration of AWF metrics in flood risk to life analysis

Real time forecasting in Scotland, as elsewhere, focuses on predicting the progress towards peak discharge and reach peak levels (using linked hydrological and hydraulic models) most often from persistent heavy rainfall causing overbank flow to risk to land and property. The AWF events described here are rarely overbank in Scotland and their peaks are rarely significant except near their source. Nevertheless, they pose a serious risk to the lives of river users exposed in or on the banks of a river from the very rapid increase in level and velocity.

There is substantial historical evidence in Scotland that the rapid rise in water levels significantly contributes to fatalities among individuals (Archer and Fowler, 2021, Archer et al., in prep). For example, in June 1835 the Caledonian Mercury reported that a man and his wife were carried away in the upper Gala Water (a tributary of the Tweed); the man was drowned but the woman was saved by being dragged by the hair to the bank. In the neighbouring River Leader three children were washed away and drowned in the same thunderstorm. British Rainfall (1882) reports that Rev MacIntyre was fishing in the Glenhinsdale River on Skye when he was carried off and drowned. He was standing along with a lad up to his knees in the water a few feet from the bank and was taken unawares by the flood; the lad had a narrow escape being carried some distance down the stream.

Water depth and velocity are generally considered the main factors in the stability of people in floodwaters (Ramsbottom et al., 2006). However, velocity per se is rarely included in flood forecasting and warning models, with discharge used as a proxy for its impact. Standard models for flood hazard assessment in the UK do not explicitly account for the additional hazard posed by rapidly rising flows which are the key to risk from AWFs. Recent hydrodynamic approaches in the literature have moved beyond static assessments and incorporate the mechanics of toppling and sliding instabilities to reflect the dynamic interaction between humans and floodwaters, particularly in rapidly varying flow conditions (Xia et al., 2014; Kvočka et al., 2018).

In Scotland, our findings suggest that the use of depth and discharge alone is insufficient for fully characterising risk to life at a national scale. The inclusion of metrics that capture the rate at which water level and flow increase offers critical additional insights. In this study, we developed and applied additional metrics to characterise the hazard associated with rapidly rates of rise.

Our analysis is based on the first derivative of level and discharge, using the full station records of 15 min flow at 260 gauging stations. The simplest of these metrics are the maximum increase in level and flow in 15 minutes (Fig 3a and b). Events are widespread but with a predominance in drier northeast Scotland and southern Scotland but with fewer observations in rivers draining the western Highlands. AWFs are suppressed by upstream lakes and reservoirs. Few are observed on upland tributaries mainly because of the paucity of gauging stations in such small catchments, but events generated in the headwaters may be transmitted downstream to catchments such as the Spey and Dee with catchment area greater than 1000 km². Normalising the absolute maximum 15 min rise in flow by QMED (the mean annual maximum peak flow (Fig 5a) provides a means of comparing the severity of AWFs between catchments of different characteristics including size. The severity of the largest

AWF on a catchment compared to the median (Fig 5b) provides a measure of the additional hazard on a catchment where extreme events do not normally occur and may be least expected. Fig 6 shows the proportional increase in flow between the start and end of the maximum 15 min rise with 23 stations showing an increase of more than 20 times, which would clearly pose a challenge to even the fittest river user. Our results show that these metrics do not spatially coincide, each metric highlights different catchment types.

Floods are a major concern for the Scottish Government and the Scottish Environment Protection Agency (SEPA), which have developed rigorous, country-wide flood forecasting methodologies and comprehensive warning service frameworks aimed at reducing risk to life (Cranston et al., 2011; Scottish Environment Protection Agency, 2022; Speight et al., 2018, 2019). We show here that an important step in the prevention of the risk to life is to account beyond traditional peak level and discharge hazard indicators, particularly when considering a national scale approach, where different hydroclimatic and geomorphologic characteristics will present different types of AWFs.

We recommend that the hazard of rapid rise in river level, velocity and discharge be given separate consideration from peak flows in monitoring, modelling and forecasting in Scotland, especially given the rising number of intense, localised extreme rainfall events from warming temperatures and the projected increases in extreme downpours with global warming (Fowler et al. 2021).

# 5.2. Intensification of AWFs downstream and implications during flood hazards







Although AWFs are generally assumed to originate in steep, upland tributaries, our analysis of two events on the Findhorn reveals that the most pronounced rates of rise occurred downstream, in lowland areas with relatively gentle slopes. This suggests that AWFs can propagate and intensify in main river channels, affecting stretches of river where flood hazard assessments are not commonly applied. This is supported by other case studies in the River South Tyne (Archer et al., 2024; Archer and Fowler, 2018) (Fig 1). the Findhorn flood of June 1990 (Fig. 8) demonstrates that shock waves can be achieved by the transformation of a normal flood wave in the main stem of a river. Several events in our analysis show a steepening and increase in the magnitude of the wave front between the upstream station at Shenachie and the downstream one at Forres and the absorption of several initial upstream waves into a single wave front at the downstream station.

The behaviour observed in Scotland and in previous examples is consistent with kinematic wave theory (Lighthill and Whitham, 1955). Our analysis supports this interpretation, providing real-world examples of such wave steepening and highlighting the need to account for these dynamics in hazard assessments. Lighthill and Whitlam (1955) note that kinematic shock waves can develop due to the overtaking of slower waves by faster ones and that they can increase in strength (magnitude of wave front level or discharge) and unite with other shock waves to form a single shock wave. However, the existence of shock waves in real rivers (as opposed to hydraulic models) has been subject to uncertainty and dispute in the past absence and rarity of real examples (Henderson, 1966; Cunge, 1969; Kibler and Woolhiser, 1970; Miller, 1984; Ponce, 1991; Beven, 2012). The Findhorn events are a practical example of how flood waves can evolve and increase with a steepening of the flood wave in downstream reaches, evolving into life-threatening events.

#### 5.3 Limitations and future directions for AWF analysis

A key limitation of this study is the rarity of gauged observations in small upland catchments where AWFs are most likely to originate. Only 6% of gauged AWFs were recorded on catchments with an area less than 50 km². However, because of the near random occurrence of AWF generating storms, a gauge placed on a given upstream tributary may not record an AWF for decades. Many stations in this study recorded only a single event in a 30 to 40 year record. General expansion of the network of headwater gauging stations is therefore unlikely to be cost effective but it may be feasible to target headwaters of catchments where multiple AWFs have been observed downstream, such as the rivers Findhorn, Spey, Cassley and Ruchill Water. As well as providing warning for vulnerable downstream river users, these sites would be used to better understand the initial creation and transformation of the wave front as it moves downstream.

Rapid increase in level and velocity/discharge contribute to hazard in AWF events. Errors are likely to be limited in level measurement which provides the initial evidence for the occurrence and severity of AWFs. The assessment of discharge in AWFs using rating curves can be more problematic. Rating curves are a known source of imprecision in hydrology (Coxon et al., 2015; Di Baldassarre et al., 2009). They are typically developed in steady flow conditions and do not account for the hysteresis effects in rapidly varying flow where the level in the rising limb of the AWF can produce a much greater flow than at the same level in steady flow conditions. Discharge measurement during these events is often impractical using traditional in-river techniques but emerging technologies using noncontact measurements to estimate river discharge (Dolcetti et al., 2022; Perks et al., 2020; Vandaele et al., 2023) offer promising solutions for observing AWF dynamics in otherwise hard-to-monitor environments.






A further limitation concerns the temporal resolution of available data. Since the 1960s, river level in the UK has typically been recorded at 15-min intervals. However, this resolution may not fully capture the dynamics of AWFs. In many cases, it is unclear whether the recorded rise occurred steadily across the interval or within a matter of seconds. Archer et al., (2023) show that wave fronts can pass a gauging station almost instantaneously. In some cases, the most rapid rise may be split across two consecutive 15-minute periods, or the true peak may pass between measurements, resulting in underestimation of both the rate and magnitude of change. To improve future observations, we recommend that selected key stations in the catchments noted above to be tested with sub-15-minute logging intervals, particularly during the summer period.

A final limitation relates to the role of debris during AWF events, which is not captured in our analysis. Historical accounts in the UK describe AWFs in steep upland catchments transporting large bedload material, including boulders over a metre in diameter, which significantly increases the hazard to river users (Carling, 1986; Watkiss and Archer, 2023). Floating debris, such as logs or vegetation, can travel much farther downstream and disrupt both flow conditions and gauging station measurements (Archer et al., 2024). While video evidence from European events, such as those in Murgang (https://www.youtube.com/watch?v=2Rfuoylv34k) and Laui Giswil (https://www.youtube.com/watch?v=ZM6Pkf5argY), Switzerland, illustrates the destructive potential of entrained debris, comparable visual records are currently lacking for the UK, despite similar events being described historically (Archer & Fowler, 2021).

## **6 Conclusions**



- 1. This study presents novel large-sample evidence of abrupt wave front floods (AWFs) in Scottish rivers, offering new perspectives on their spatial distribution, downstream evolution, and hazard potential. The rapidity of onset of AWFs, often as a visible wave, provides a critical danger to the lives of river users even when the peak river level is not severe.
- 2. Mapped metrics of extreme rise in river level and discharge show that events are widespread including on catchments greater than 1000 km2 in area. Their observation at downstream locations indicates that wave fronts persist from a usual source in a headwater tributary through a long river reach.
- 3. The severity and threat to life of AWFs is illustrated by 15 min increases in level of more than 1.4 m and/or discharge of over 100 m3 s-1 at 12 stations in Scotland.
  - 4. Further metrics of discharge illustrate different aspects of risk. The ratio of the absolute maximum QWabs to the median maximum QWmed 15 min change in discharge shows how severe the most extreme event was on a given catchment. Normalising the absolute maximum 15 min rise in flow by QMED (the mean annual maximum peak flow) provides a means of comparing the severity of AWFs between catchments of different characteristics including size.
  - 5. The magnitude of the wave front, expressed as a multiple of the initial flow, provides another measure of the AWF hazard. We found that 11 stations had discharges rising by more than 30 times over the 15 min interval.
  - 6. Examples of flood wave transformation including steepening on the River Findhorn provide further evidence of real-world kinematic shock generation and transmission.
- 7. We recommend that the hazard of AWFs needs to be given separate consideration from peak flows in monitoring, modelling and forecasting in Scotland, especially given the rising number of intense, localised extreme rainfall events from warming temperatures and the projected increases in extreme downpours with global warming (Fowler et al. 2021

#### Acknowledgements

This work is funded by the Natural Environment Research Council (NERC)-sponsored ONE Planet Doctoral Training Partnership- grant reference number: NE/S007512/.

#### **Competing interests**

The authors declare that they have no conflict of interest.

#### **Author contributions**

- D R Archer: conceptualisation, methodology, formal analysis, writing draft and review
  - F Fileni: data curation, software, formal analysis, methodology, writing review and editing
  - S. A. Watkiss: investigation, visualisation
  - H. J. Fowler: project administration, supervision, writing review and editing

## 425 Data and code availability

The 15 min flow and level data used in this study is available at the SEPA Time Series Data Service (API) (https://timeseriesdoc.sepa.org.uk/) the code and rates of rise data extracted from the timeseries may be accessed by other researchers: https://doi.org/10.5281/zenodo.14771542 or via GitHub repository for the latest version.

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
