# Peer review of "Characteristics of gauged abrupt wave fronts (walls of water) in flash floods in Scotland"

_EGUsphere, 2025_

## Author Response (AR1)

| Nbr | Comment                                                                                    | Response                                                   |
|-----|--------------------------------------------------------------------------------------------|------------------------------------------------------------|
|     | The changes suggested by CC1 are largely cosmetic, but they involve changing the format of |                                                            |
|     | the principal variables. This means changes in nearly every paragraph. We have prepared a  |                                                            |
|     | file comparing the original submission with the revised submission and have also untracked |                                                            |
| _   | this to provide a clean version of the paper.                                              |                                                            |
| 1   | It is good to see that the authors' work on                                                | No response needed                                         |
|     | this important, previously-neglected                                                       |                                                            |
|     | hydrological hazard has been extended into Scotland. I offer just a few                    |                                                            |
|     | comments from a quick look the paper.                                                      |                                                            |
| 2   | The use of Q15 to mean annual maximum                                                      | Accepted that terminology is a potential source            |
|     | rise in discharge over 15-minutes is a                                                     | of confusion. To avoid the conflict of use of Q15          |
|     | potential cause of confusion (especially                                                   | for flow duration we have changed all                      |
|     | for any readers who, like I did, start from                                                | references to QW15 as the 15-minute wave                   |
|     | the conclusions and work backwards).                                                       | increase in discharge. This involves numerous              |
|     | Q15 is commonly used in hydrology to                                                       | changes through the text.                                  |
|     | refer to the 15th percentile on a flow                                                     | For consistency we have changed H15 to HW15                |
|     | duration curve. I suggest a change in                                                      | ,                                                          |
|     | terminology.                                                                               |                                                            |
| 3   | Also I'd suggest rephrasing " annual                                                       | Accepted and changed                                       |
|     | maximum values of rise in level and                                                        | -                                                          |
|     | discharge" to "annual maximum values of                                                    |                                                            |
|     | rise in level and rise in discharge", to                                                   |                                                            |
|     | avoid any misunderstanding.                                                                |                                                            |
| 4   | There are several other instances where                                                    | Accepted and altered to:                                   |
|     | some rephrasing could aid clarity such as                                                  | Coincidence between level and flow station                 |
|     | "coincidence between level and flow                                                        | maximum rates of rise, where the maximum                   |
|     | station maxima" which I believe is                                                         | H15 exceeded 0.6 m, occurred for 48% of                    |
|     | intended to refer to maximum rates of                                                      | stations                                                   |
| 5   | rise.                                                                                      | We have applied the median rather than mean                |
| 5   | Likewise, does the "the mean maximum 15 min rise " refer to the mean of the                | throughout our analysis (corrected).                       |
|     | annual maxima? And does the median in                                                      | For further clarification we have used QW15 med |
|     | section 3.1 refer to the median of the                                                     | for the median of the annual maximum                       |
|     | annual maxima? Later, this is called                                                       | increase in 15-minute discharge and QW15 abs    |
|     | RORMED.                                                                                    | for the absolute maximum 15-minute increase                |
|     |                                                                                            | in discharge.                                              |
| 6   | In 3.2, the authors should state that c                                                    | Accepted and added                                         |
|     | refers to celerity, and give the units of all                                              |                                                            |
|     | variables.                                                                                 |                                                            |
| 7   | I cannot see that Eqn 1 is consistent with                                                 | We have modified the equations and                         |
|     | the definition given above. For one thing,                                                 | terminology to distinguish between median                  |
|     | it has no expression of maximisation.                                                      | annual and absolute maximum 15-min                         |
|     |                                                                                            | maximum increase in 15 minutes                             |
| 8   | It seems unfortunate that the first                                                        | We think it is a reasonable approach to use our            |
|     | examples presented, in section 1, are all                                                  | previous research in an adjacent area as a                 |
|     | in the north of England rather than in                                                     | guide to the characteristics of the events which           |
|     | Scotland, given the title of the paper.                                                    | we are likely to experience in Scotland.                   |
|     |                                                                                            | However, for clarification we have added:                  |

| 'We use the lessons learned from this analysis |
|------------------------------------------------|
| in the extension here to neighbouring Scotland |
| with a greater range of mountain               |
| environments'                                  |

RC1: 'Comment on egusphere-2025-456', Charlie Pilling, 18 Apr 2025

| No | Comment                                     | Response                                   |
|----|---------------------------------------------|--------------------------------------------|
| 1  | This is a well written, balanced paper that | We thank the reviewer for that encouraging |
|    | considers an important hazard that can      | comment.                                   |
|    | be overlooked. It has been extensively      |                                            |
|    | researched and presented with thorough      |                                            |
|    | analysis. This is a valuable contribution   |                                            |
|    | and recommend that it is accepted 'as is'   |                                            |

RC2: 'Comment on egusphere-2025-456', Anonymous Referee #2, 14 Jun 2025 reply

| Nhr    | Comment by reviewer                                                                         | Posnonso                                                      |
|--------|---------------------------------------------------------------------------------------------|---------------------------------------------------------------|
| Nbr    | Comment by reviewer                                                                         | Response                                                      |
|        | In order to strengthen the paper and in response to issues raised by RC2, we have rewritten |                                                               |
|        | the Introduction and Discussion to put the research into a better context with flash flood  |                                                               |
|        | research generally. We have retained the substance of Sections on Data, Methods and         |                                                               |
|        |                                                                                             | viewer comments as listed below                               |
| 1      | The paper presents an analysis                                                              | As stated in the paper, the dataset was sourced from SEPA,    |
|        | of a data set from the Scottish                                                             | and as such it is not new. The aim of the paper was to        |
|        | environmental protection                                                                    | derive metrics from the existing data. We have created a      |
|        | agency. This is not a new data                                                              | new dataset of annual and absolute maximum rates of rise      |
|        | set.                                                                                        | for the 260 gauging stations which provide the basis for      |
|        |                                                                                             | the analysis in the paper. The novelty factor stands in using |
|        |                                                                                             | these created metrics to address the phenomenon of rates      |
|        |                                                                                             | of rise and the associated risk to life, a phenomenon rarely  |
|        |                                                                                             | addressed by researchers. Given that there are more than      |
|        |                                                                                             | 20.000 years of continuously measured 15-min data across      |
|        |                                                                                             | the 260 stations, these metrics were a necessity to           |
|        |                                                                                             | summarize the issue and derive the conclusions.               |
| 2      | The analysis are very limited                                                               | The reviewer does not address the purpose of the paper        |
|        | and naïve. There are some                                                                   | which is to highlight the risk to life of very rapid rates of |
|        | basic flaws. The novelty and                                                                | rising level and discharge rather than the focus on peak      |
|        | scholarship are thin.                                                                       | flow typical of flood risk analysis including flash floods.   |
|        |                                                                                             | We accept that the concepts are simple, but they are not      |
|        |                                                                                             | naïve. The analysis is novel in that few papers worldwide     |
|        |                                                                                             | have addressed this aspect of flood risk. Hence, the          |
|        |                                                                                             | predominance of References by the authors.                    |
|        |                                                                                             | By scholarship, we wonder whether the reviewer means          |
|        |                                                                                             | complex? We did not intend to address the hydraulic           |
|        |                                                                                             | theory of rapid rise and hydraulic shock which is not         |
|        |                                                                                             | necessary for this study.                                     |
|        |                                                                                             |                                                               |
| 3      | There is no discussion of the                                                               | To clarify our use of rated discharge data we have added      |
|        | errors and uncertainties of the                                                             | the following:                                                |
|        | analysed data! This is                                                                      |                                                               |
| $\Box$ | ,                                                                                           |                                                               |

extraordinary and it would be surprising if the information is not provided by the Scottish environmental protection agency.

There are plenty of references of discharge and flow rate "data" which are presented without clear explanation if these were measured discharges or estimated discharges based upon rating curves. I believe that the gauging stations reported water level data only and that some 'rating curves' were applied.

'The data provided by SEPA are based on 15-min measurements of river level and these are with few exceptions converted to flow by rating equations derived from individual discharge measurements at given levels combined with weir equations. SEPA hydrometry team reviews level measurements monthly and rating curves annually. These reviews result in the correction of data artefacts before publishing the timeseries and necessary changes in the rating curves and flow conversions. We further visually inspected all hydrographs of rapid rates of rise and eliminated spurious records following a comprehensive QC procedure (Fileni et al. 2023)'.

We give further examples of published papers which highlight the importance of 'rated' data in large sample studies. The use of an ample number of stations and measurements with a duration of several years allows conclusions to be drawn at large spatial and temporal scale for instance:

Xuan Do et al., 2020 (HESS):Studies the trend on extreme flows across 3666 river gauges from 1971 to 2005. Iliopoulou et al., 2019 (HESS):Study uses 224 rivers with more than 50 years of data each drawing conclusions of seasonal patterns at a continental scale Slater et al., 2021 (GRL):Uses 10093 gauge records with information from the before the 80s until the 2000s to access non stationarity of high return periods in flows

In our study, we use 260 stations from Scotland, commonly used in both scientific (Lane et al., 2019, 2022; Lees et al., 2021) and planning and regulation applications in the UK (Wallingford HydroSolution, 2019) The conclusions that we draw would not have been possible without this data.

discharge rating curves of streams and rivers are typically developed based upon steady flow conditions and assumptions. But It is well-known that the rating curve differs at a given site between the rising hydrograph and declining hydrograph, for the same water depths. The differences increases inversely proportional to the duration of the flood event, with major hysteresis during flash floods.

We are well aware of hysteresis in rating curves and the impact of rapidly changing levels on the equivalent discharges. However, the basis for this research is the observation of such rapid rates of rise in level as to cause a risk to life of river users. The impact of hysteresis on discharge assessment is not critical. However, on the basis of the reviewer's comment we have considered the implications of hysteresis and rated flow measurements in the revised Discussion section.

5 The bibliographic review is poor

We agree that there are numerous papers addressing aspects of flash floods, either in river or in surface water. However, as noted above, there are few addressing

|   | A number of relevant literature on flash floods were ignored, including UK studies. The list is too long to develop and a very limited number of recent works are listed below.                                                                                                                                         | specifically the impact of observed rates of rise which this research shows that in Scotland can reach nearly 2 m rise in the 15-min observation interval. We have strengthened the context of our research with further references in Introduction and Discussion  We thank the Reviewer for the list of flash flood papers and consider them individually in their relevance to this research.                                                                                                                                                                                                 |
|---|-------------------------------------------------------------------------------------------------------------------------------------------------------------------------------------------------------------------------------------------------------------------------------------------------------------------------|--------------------------------------------------------------------------------------------------------------------------------------------------------------------------------------------------------------------------------------------------------------------------------------------------------------------------------------------------------------------------------------------------------------------------------------------------------------------------------------------------------------------------------------------------------------------------------------------------|
| 6 | HALFI, E., PAZ, D., STARK, K., YOGEV, U., REID, I., DORMAN, M., and LARONNE, J.B. (2020). "Novel mass-aggregation-based calibration of an acoustic method of monitoring bedload flux by infrequent desert flash floods." Earth Surface Processes and Landforms, Vol. 45, No. 14, pp. 3510-3524 (DOI: 10.1002/esp.4988). | The focus of this paper is on calibration of an acoustic method of monitoring bedload flux with accompanying field measurements of level and estimated discharge on a limited number of infrequent desert floods. Field data acquisition for such research differs from the extraction of critical events from an archive of existing level and flow data. We cannot see any relevance to the occurrence of rapid rates of rise in level and discharge and therefore we do not think it is relevant to our paper.                                                                                |
| 7 | HALFI, E., THAPPETA, S.K., JOHNSON, J.P.L., REID, I., and LARONNE, J.B. (2023). "Transient bedload transport during flashflood bores in a desert gravel-bed channel." Water Resources Research, Vol. 59, Paper e2022WR033754, 17 pages (DOI: 10.1029/2022WR033754).                                                     | Here, the authors investigate how bedload transport rates change during the passage of natural flash-flood bores. We found this paper very interesting, notably with respect to the measurements of very rapid rise in level at the bore front on a dry channel and the associated rise in surface water slope and bedload transport (and declining afterwards). As above, it refers to field measurements in individual events and it is not clear how the paper is relevant to the characteristics and geographical distribution of rapid rates of rise in level from archived level and flow. |
| 8 | KVOCKA, D., FALCONER, R.A., and BRAY, M. (2015).  "Appropriate model use for predicting elevations and inundation extent for extreme flood events." Natural Hazards, Vol. 79, pp. 1791-1808 (DOI: 10.1007/s11069-015-1926-0).                                                                                           | This paper discusses appropriate models for assessing peak water levels and inundation extent and concludes that methods including shock capturing are necessary for steep catchments presumably affected by intense rainfall causing flash flooding. The paper does not discuss the modelling of the rate of rise of the wave front which presumably would also require the inclusion of shock capturing. Though interesting, we do not consider the paper relevant to our analysis.                                                                                                            |
| 9 | KVOCKA, D., AHMADIAN, R., and FALCONER, R.A. (2018). "Predicting Flood Hazard Indices in Torrential or Flashy River Basins and Catchments." Water Resources Management, Vol. 32, pp. 2335-2352 (DOI: 10.1007/s11269-018-1932-6).                                                                                        | Given our primary interest in the risk to river users of rapidly rising flood levels, this is an interesting paper on assessing human stability in floodwaters. We have added reference to it in the paper. (It is of more interest for our next paper in which we map and discuss the characteristics of catchments in which rapid rates of rise (AWFs) occur. This provides alternative methodologies).                                                                                                                                                                                        |

| 10 | The Introduction is very poorly developed. It is self-centered around the authors' publications with self-citations after self-citations.                                                                                                                                                                                                                                                                                                                                                                                            | This is true but it is due to the very limited work done on rapid rates of rise by other authors.                                                                                       |
|----|--------------------------------------------------------------------------------------------------------------------------------------------------------------------------------------------------------------------------------------------------------------------------------------------------------------------------------------------------------------------------------------------------------------------------------------------------------------------------------------------------------------------------------------|-----------------------------------------------------------------------------------------------------------------------------------------------------------------------------------------|
| 11 | The text includes some quantitative discharge numbers without explanations if these were measured discharges or estimated discharges based upon rating curves                                                                                                                                                                                                                                                                                                                                                                        | We have added a to the Data section to explain the source of the discharge numbers, noted in Response 3. We did not include this in our original text as is standard practice for SEPA) |
| 12 | Line 2 suggests 20,000 years of data. Truly amazing, if true, but most likely a typographic mistake.                                                                                                                                                                                                                                                                                                                                                                                                                                 | This is true as an aggregate across 260 stations as explained in Response 1                                                                                                             |
| 13 | There is no discussion of the errors and uncertainties of the analysed data! This is extraordinary and it would be surprising if the information is not provided by the Scottish environmental protection agency.  There are plenty of references of discharge and flow rate "data" which are presented without clear explanation if these were measured discharges or estimated discharges based upon rating curves. I believe that the gauging stations reported water level data only and that some 'rating curves' were applied. | We have added some sentences into the paper on how the data was QCd. Also see above comment in Response 3                                                                               |
|    | Any mention of discharge and flow rate should explicitly state 'measured' or 'rated', with the implicit limitations of the latter                                                                                                                                                                                                                                                                                                                                                                                                    | We have put this detail into the data and methods section.                                                                                                                              |
| 14 | On '3.2 Change in velocity': the authors state "Initial velocity before of the arrival of the AWP [] is likely to be dominated by wave celerity". This is incorrect and untrue.                                                                                                                                                                                                                                                                                                                                                      | The reference to 'domination by wave celerity' does not refer to initial conditions before the arrival of the AWF but during the AWF. We have added text to clarify the distinction.    |
|    | In most streams and rivers, the initial velocity will be close to the uniform equilibrium                                                                                                                                                                                                                                                                                                                                                                                                                                            |                                                                                                                                                                                         |

|    | velocity, also called 'normal' velocity, derived from momentum considerations. That is, the longitudinal slope of the water surface would be very close to or equal to the bed slope, and the velocity would fullfill the equilibrium between the gravity force component in the flow direction and the boundary shear force resisting the flow motion.                                                                                                                                                                                                                                                                                                                                                                 |                                                                                                                                                                                  |
|----|-------------------------------------------------------------------------------------------------------------------------------------------------------------------------------------------------------------------------------------------------------------------------------------------------------------------------------------------------------------------------------------------------------------------------------------------------------------------------------------------------------------------------------------------------------------------------------------------------------------------------------------------------------------------------------------------------------------------------|----------------------------------------------------------------------------------------------------------------------------------------------------------------------------------|
| 15 | On '3.2 Change in velocity': the 'basic equation c=dQ/dA' is the celerity of a monoclinal wave. The monoclinal wave is a mathematical approximation assuming steady flow conditions before and after the flood front. The approximation does not apply to flash flood.                                                                                                                                                                                                                                                                                                                                                                                                                                                  | WE did not find this method practical for estimating celerity for an AWF flash flood so we accept your criticism and have deleted this paragraph.                                |
| 16 | There are plenty of references of discharge "data" which are presented without clear explanation if these were measured discharges or estimated discharges based upon rating curves. I believe that the gauging stations reported water level data only and that some 'rating curves' were applied.  the discharge rating curves of streams and rivers are typically developed based upon steady flow conditions and assumptions. But It is well-known that the rating curve differs at a given site between the rising hydrograph and declining hydrograph, for the same water depths. The differences increases inversely proportional to the duration of the flood event, with major hysteresis during flash floods. | This is now detailed in the data and methods section with details in Responses 3 and 4.  An appropriate reflection of the issues of hysteresis is now included in the Discussion |

| 17 | The section must be drastically |
|----|---------------------------------|
|    | restructured and rewritten,     |
|    | with removal of flawed data     |
|    | interpretation.                 |
|    | All the sub-sections related to |
|    | some interpretation of          |
|    | discharges should be removed:   |
|    | That is, sub-sections 4.3, 4.4, |
|    | 4.5.                            |

We do not agree with this statement – this flow data is standard flow data used in the UK – we have added more explanation of where this comes from in Responses 1 and 3 and in the paper.

**References**

Iliopoulou, T., Aguilar, C., Arheimer, B., Bermúdez, M., Bezak, N., Ficchì, A., Koutsoyiannis, D., Parajka, J., Polo, M. J., Thirel, G., and Montanari, A.: A large sample analysis of European rivers on seasonal river flow correlation and its physical drivers, Hydrol Earth Syst Sci, 23, 73–91, https://doi.org/10.5194/hess-23-73-2019, 2019.

Lane, R. A., Coxon, G., Freer, J. E., Wagener, T., Johnes, P. J., Bloomfield, J. P., Greene, S., Macleod, C. J. A., and Reaney, S. M.: Benchmarking the predictive capability of hydrological models for river flow and flood peak predictions across over 1000 catchments in Great Britain, Hydrol Earth Syst Sci, 23, 4011–4032, https://doi.org/10.5194/hess-23-4011-2019, 2019.

Lane, R. A., Coxon, G., Freer, J., Seibert, J., and Wagener, T.: A large-sample investigation into uncertain climate change impacts on high flows across Great Britain, Hydrol Earth Syst Sci, 26, 5535–5554, https://doi.org/10.5194/hess-26-5535-2022, 2022.

Lees, T., Buechel, M., Anderson, B., Slater, L., Reece, S., Coxon, G., and Dadson, S. J.: Benchmarking data-driven rainfall-runoff models in Great Britain: A comparison of long short-term memory (LSTM)-based models with four lumped conceptual models, Hydrol Earth Syst Sci, 25, 5517–5534, https://doi.org/10.5194/hess-25-5517-2021, 2021.

Slater, L., Villarini, G., Archfield, S., Faulkner, D., Lamb, R., Khouakhi, A., and Yin, J.: Global Changes in 20-Year, 50-Year, and 100-Year River Floods, Geophys Res Lett, 48, https://doi.org/10.1029/2020GL091824, 2021.

Wallingford HydroSolution: ReFH2 Science Report Deriving ReFH catchment based parameter datasets in Scotland, Wallingford, 2019.

Xuan Do, H., Zhao, F., Westra, S., Leonard, M., Gudmundsson, L., Eric Stanislas Boulange, J., Chang, J., Ciais, P., Gerten, D., Gosling, S. N., Müller Schmied, H., Stacke, T., Telteu, C. E., and Wada, Y.: Historical and future changes in global flood magnitude - evidence from a model-observation investigation, Hydrol Earth Syst Sci, 24, 1543–1564, https://doi.org/10.5194/hess-24-1543-2020, 2020.